# Risk Factors for Stent Migration into the Abdominal Cavity after Endoscopic Ultrasound-Guided Hepaticogastrostomy

**DOI:** 10.3390/jcm10143111

**Published:** 2021-07-14

**Authors:** Kazushige Ochiai, Toshio Fujisawa, Shigeto Ishii, Akinori Suzuki, Hiroaki Saito, Yusuke Takasaki, Mako Ushio, Sho Takahashi, Wataru Yamagata, Ko Tomishima, Tadakazu Hisamatsu, Hiroyuki Isayama

**Affiliations:** 1Department of Gastroenterology, Graduate School of Medicine, Juntendo University, Tokyo 113-8421, Japan; kaz.ochiai.0728@gmail.com (K.O.); t-fujisawa@juntendo.ac.jp (T.F.); sishii@juntendo.ac.jp (S.I.); suzukia@juntendo.ac.jp (A.S.); hiloaki@juntendo.ac.jp (H.S.); ytakasa@juntendo.ac.jp (Y.T.); m-ushio@juntendo.ac.jp (M.U.); sho-takahashi@juntendo.ac.jp (S.T.); w.yamagata.mx@juntendo.ac.jp (W.Y.); tomishim@juntendo.ac.jp (K.T.); 2Department of Gastroenterology and Hepatology, Kyorin University School of Medicine, Tokyo 192-8508, Japan; thisamatsu@ks.kyorin-u.ac.jp

**Keywords:** obstructive jaundice, interventional EUS, EUS-guided biliary drainage (EUS-BD), EUS-guided hepaticogastrostomy (EUS-HGS), stent migration

## Abstract

**Background:** Endoscopic ultrasound-guided hepaticogastrostomy (EUS-HGS) is becoming increasingly popular. However, the risk factors for stent migration into the abdominal cavity remain unknown. **Methods:** Forty-eight patients undergoing EUS-HGS with placement of a long, partially covered self-expandable metallic stent (LPC-SEMS) were studied retrospectively to identify risk factors of stent migration. We determined the technical and functional success rates, and recorded adverse events, including stent migration. **Results:** EUS-HGS was technically successful in all patients. However, stent migration was evident in five patients (one actual and four imminent, 10%). Stent migration into the abdominal cavity was observed in one patient (2%), and the other four cases required additional procedures to prevent migration (8%). Logistic regression analysis revealed that the risk of stent migration increased as the initial (pre-procedure) distance between the stomach and liver at the puncture site increased (*p* = 0.012). **Conclusions:** A longer distance between the stomach and liver at the puncture site increased the risk of stent migration. However, during EUS-HGS, it is difficult to adjust the puncture position. It is important to ensure that the proportion of the stent in the stomach is large; the use of a self-anchoring stent may be optimal.

## 1. Introduction

Endoscopic ultrasound-guided hepaticogastrostomy (EUS-HGS) is increasingly being used in cases where endoscopic retrograde cholangiopancreatography (ERCP) fails, or is difficult [1] because of altered gastrointestinal anatomy, gastric outlet obstruction (GOO), or the presence of a duodenal stent [2,3,4] EUS-HGS is safer than percutaneous transhepatic biliary drainage (PTBD) [5,6]. However, as there is no dedicated device for EUS-HGS, in contrast to ERCP and PTBD, sometimes, this can be difficult [7]. Possible complications of EUS-HGS include bleeding, peritonitis, mediastinitis and stent displacement/migration. Stent migration into the abdominal cavity is a serious complication that often requires surgery because it causes severe peritonitis, [8,9,10,11,12] although some stents have been recovered endoscopically [13]. Other complications include an increased distance between the liver and gastric wall (revealing the Candy sign on fluoroscopy) and the opening proximal end of the stent in the abdominal cavity. In such cases, another stent can be placed in tandem to connect the intraabdominal stent to the stomach, over the guidewire passed through the deployed stent to prevent bile leakage into the peritoneal cavity [14]. The incidence of stent misplacement has recently decreased given the remarkable advances in intra-channel deployment and EUS-HGS techniques [15]. However, a few days after successful stent placement, stomach peristalsis may cause the stent to migrate into the abdominal cavity. The clipflap [16] and crisscross [17] methods prevent migration after successful stenting, but the procedure times are longer. The factors predicting stent migration after EUS-HGS remain unknown, and we aimed to identify them in this study.

## 2. Methods

### 2.1. Patients

Consecutive patients with unresectable malignant biliary obstruction (MBO), who underwent EUS-HGS using long, partially covered, self-expandable metallic stents (LPC-SEMS) between March 2017 and April 2020 at Juntendo University Hospital, were enrolled retrospectively. The inclusion criteria were EUS-HGS with LPC-SEMS placement to treat unresectable MBO of various etiologies. The exclusion criterion was use of a plastic stent for EUS-HGS.

### 2.2. The Stents and the Procedures of EUS-HGS

EUS-HGS was performed by 6 well-trained endoscopists with one supervisor, who experienced more than 100 EUS-HGS procedures. The intrahepatic bile duct was punctured with a 19G fine-needle aspiration (FNA) needle. Then, contrast medium was injected to delineate the intrahepatic bile duct. Once an appropriate cholangiogram was obtained, a 0.025-inch guidewire was inserted into the intrahepatic bile duct. Next, both the bile duct and the stomach wall were dilated using a bougie dilater and/or 4-mm balloon catheter. Finally, we placed an LPC-SEMS (Niti-S S-type stent, Taewoong Medical Co. Ltd., Gimpo, Korea), which is made of braided nitinol wire and partially covered inside with a silicone membrane. The hepatic end has a 1-cm uncovered portion, and the gastric end has a flared portion. The stents used in this study were 8 or 10 mm in diameter and 10 or 12 cm in length. The stent length was selected to ensure the intragastric portion of the stent was longer than 5 cm. When we placed the stents, we used the intra-channel release technique for all EUS-HGS cases.

### 2.3. Study Outcomes

We aimed to identify factors predisposing patients to stent migration into the abdominal cavity.

### 2.4. Definitions

We performed computed tomography (CT) before EUS-HGS and on the day after the procedure. We measured the distance between the puncture site and the stomach on both scans (Figure 1). The distance from the cardia to the stent was measured on the post-procedure scans.

We defined technical success as appropriate placement of an LPC-SEMS between the intrahepatic bile duct and the stomach. Stent migration was defined as migration into the abdominal cavity or the need for additional treatment to avoid migration, as revealed by postoperative CT (Figure 2 and Figure 3). If the length of stent in the the stomach was too short, we performed additional procedures. Functional success was defined as a reduction in the serum total bilirubin level by 50% or to <2 mg/dL within 2-week figures. Adverse events were defined as described previously [18]. Peritonitis was diagnosed on the basis of clinical peritoneal inflammation.

### 2.5. Statistical Analyses

Data are presented as numbers with percentages or medians with an interquartile range (IQR). Fisher’s exact test and Student’s *t*-test were used to compare categorical and continuous variables between groups, respectively. Factors significant at *p* ≤ 0.1 in univariate analysis were included in the multivariate analysis. All statistical analyses were performed using SPSS software (ver. 19.0; SPSS Inc., Chicago, IL, USA).

## 3. Results

### 3.1. Patients

Between March 2017 and April 2020, 48 procedures were performed on 47 patients (mean age, 71 years; 63% males). All procedures involved EUS-HGS with LPC-SEMS placement to treat MBO (Table 1). The primary cause of MBO was pancreatic cancer in 24 cases (50%). Of the total 48 procedures, the major reason for EUS-HGS was GOO (56% of cases); the other reasons were a surgically altered anatomy (21%), failed ERCP (21%), and the decision not to perform high-risk ERCP (2%). Sixteen patients (33%) underwent prior biliary drainage and duodenal stents were used in 10 (21%) cases. Table 2 shows the details of the procedure. The median procedure time was 42 min (IQR: 29–55 min). We punctured segment 3 of the bile duct (B3) in 41 patients (85%) and segment 2 (B2) in 7 (15%). In total, 45 patients (94%) underwent fistular dilation using a bougie, 40 (83%) underwent balloon dilation, and in 5 (10%) dilation was achieved via cautery. The stent diameter was 8 mm in 45 patients (94%) and 10 mm in 3 (6%). The stent length was 10 cm in 42 patients (88%) and 12 cm in 6 (13%).

### 3.2. Technical and Clinical Success, and Adverse Events

Table 3 shows the technical and clinical success rates, and the rates of adverse events including stent migration. EUS-HGS was technically successful in all patients. The clinical success rate was 90%; adverse events were observed in 8 cases (17%). The adverse events included stent migration (10%) and peritonitis (6%). The stent migrated into the abdominal cavity in 1 case (2%), and 4 cases required additional procedures to prevent migration (8%).

### 3.3. Stent Migration

We aimed to identify the risk factors for LPC-SEMS migration. Table 4 shows that the non-migration (43 cases) and migration (5 cases including 1 actual and 4 imminent) groups did not differ in terms of age, sex, performance score, or pancreatic cancer, liver metastasis, ascites, or distal stricture status. All five cases in the migration group need additional anti-migration or recovery procedure. The groups also showed no difference in the proportion of cases where EUS-HGS was caused by GOO, or in prior biliary drainage or duodenal stenting status. The initial distance between the puncture site and the stomach was associated with the risk of stent migration (*p* ≤ 0.001), while the initial distance from the cardia to the stent showed a trend toward an association with the risk of stent migration (*p* = 0.097). Logistic regression analysis revealed that the risk of stent migration increased as the initial distance between the puncture site and the stomach increased (*p* = 0.012).

We also compared the stomach-to-liver distance before and after the procedure. The distance was shorter after the procedure compared to before the procedure, but not significantly, in both the migration and non-migration groups (*p* = 0.691, *p* = 0.291 respectively) (Table 5).

## 4. Discussion

A long distance between the stomach and liver before stent placement increased the risk of stent migration. The post-procedure distance was shorter, but not significantly (Table 5). Thus, when a SEMS is placed during EUS-HGS, the stomach will return to its original position. The scope is pressed against the stomach wall during EUS-HGS, so the stomach and liver are in close contact. The distance between the stomach and liver can be minimised using an intra-channel SEMS, with the stomach and liver pressed together [15]. We expected that the SEMS would keep the stomach and liver close together after EUS-HGS. However, the stomach-to-liver distances before and after EUS-HGS were 19.1 ± 12.8 and 17.7 ± 9.8 mm (*p* = 0.291), respectively, in the non-migration group, and 47.0 ± 11.3 and 44.8 ± 10.7 mm (*p* = 0.691), respectively, in the migration group; i.e., the distance decreased only slightly after the procedure.

A greater distance between the cardia and the puncture site was associated with a non-significantly higher risk of stent migration, because the distance between the stomach and liver is also greater. Thus, puncture of B2 at a point reasonably close to the cardia, and puncture of B3 as close to the cardia as possible, may reduce the likelihood of stent migration. However, B2 puncture is associated with a risk of esophageal puncture and mediastinitis [19]. When B3 is punctured within the vicinity of the cardia, the puncture angle is close to that of the vertical component of the peripheral small bile duct, so it may be difficult to advance the guidewire to the hilar side. Furthermore, placement of a large-bore SEMS is difficult. In addition, if the stent is placed directly under the esophagogastric junction, the passage of food is compromised, which is problematic because re-intervention to deal with stent occlusion is not easy.

Given the need for a guidewire, large-bore SEMS, and re-intervention in cases of difficult stent delivery, it would be preferable to puncture a location that is relatively remote from the cardia. However, this increases the risk of stent migration because of the greater distance between the stomach and liver. To prevent stent migration, it may be necessary to ensure that more of the stent is in the stomach than was the case in this study. No stent migration was noted when the median length of the portion of the stent in the stomach was 52 mm [20,21]. In addition, SEMSs with anchors may not migrate. One such stent (the Spring Stopper Stent; Taewoong Medical Co. Ltd.) is under evaluation at our institution. However, if the stomach anchor is too strong, the hepatic side of the stent may be diverted into the peritoneal cavity. Thus, even if the anchors are appropriate, a longer stent is required. It is also important to develop stents that do not increase the distance between the stomach and liver.

It is difficult to determine the risk of stent migration prior to EUS-HGS. CT prior to EUS-HGS reveals the approximate distance between the stomach and liver, but not the puncture site. It is difficult to determine the distance from the cardia to the puncture site, and from the stomach to liver, during EUS-guided procedures. If post-procedural CT indicates a high risk of stent migration, this can be reliably prevented using the clip-flap or crisscross method [16,17]. When pre-operative CT clearly indicates a long distance between the stomach and liver, it may be necessary to give up EUS-HGS and treat this with alternative methods, such as percutaneous transhepatic biliary drainage, if there is a high possibility of stent migration.

The limitations of the present study included its retrospective design, the small number of cases, and the difficulty associated with accurate measurement of the stomach-to-liver distances on CT scans. Strengths included routine performance of CT on the day after EUS-HGS, the consistency associated with all data being from a single institution, and all patients receiving the same stent.

## 5. Conclusions

Multivariate analysis revealed that the risk of (actual or imminent) stent migration increased as the distance between the stomach and liver increased at the puncture site. The initial distances for the migration and non-migration groups were 47.0 ± 11.3 and 20.1 ± 12.8 mm, respectively (*p* < 0.001). Post-procedural CT revealed that the gastric wall returned to its original position. However, it is difficult to adjust the puncture position. It is important to ensure that the portion of the stent in the stomach is sufficiently long, and it may be beneficial to use a stent with an anchor at each end to prevent migration.

## Figures and Tables

**Figure 1 jcm-10-03111-f001:**
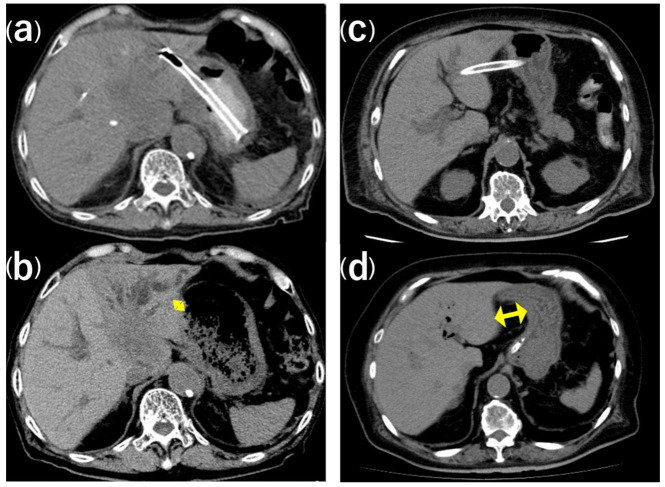
The distance between the puncture site and stomach, as revealed by CT before and after EUS-HGS. (**a**) In the non-migration group, the post-EUS-HGS distance between the puncture site and stomach was short. (**b**) The double yellow arrow denotes the short initial distance between the puncture site and stomach in the non-migration group. (**c**) In the stent migration group, the post-EUS-HGS distance between the puncture site and stomach was long. (**d**) In the stent migration group, the distance between the puncture site and stomach was significantly longer than in the non-migration group.

**Figure 2 jcm-10-03111-f002:**
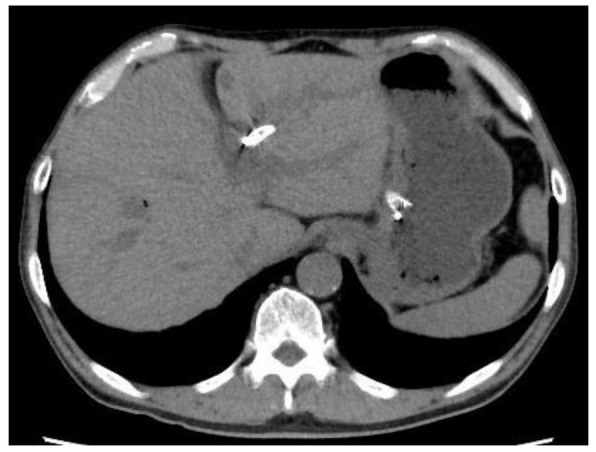
CT (Computed tomography) performed on the day after EUS-HGS revealed that the stomach stent was too short.

**Figure 3 jcm-10-03111-f003:**
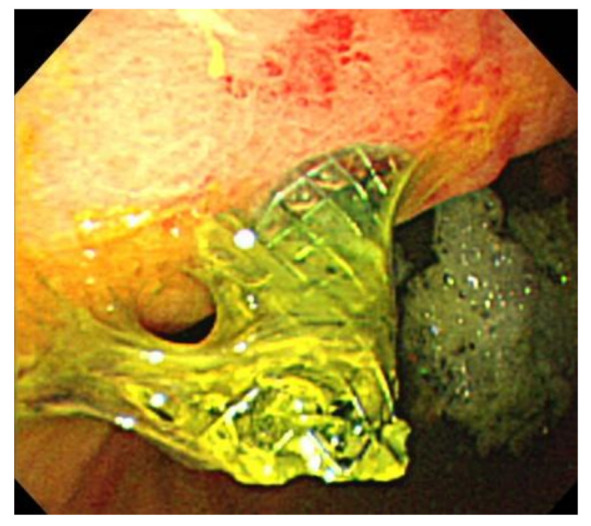
An endoscopic image of the patient shown in Figure 2. The stomach stent is very short and is about to fall out.

**Table 1 jcm-10-03111-t001:** Patient characteristics.

Age, years		71 (50–93)
Sex	Male	30 (63%)
Performance status	0/1/2/3/4	10(21%)/20(42%)/12(25%)/3(6%)/3(6%)
Primary cancer	Pancreatic cancer	24 (50%)
	Biliary tract cancer	8 (17%)
	Gallbladder cancer	2 (4%)
	Gastric cancer	4 (8%)
	Hepatocellular carcinoma	2 (4%)
	Other	8 (16%)
Liver metastasis		17 (35%)
Ascites		18 (38%)
Stricture location	Distal	39 (81%)
	Hilar	9 (19%)
Reasons for EUS-BD	GOO	27 (56%)
	Altered anatomy	10 (21%)
	Failed ERCP	10 (21%)
	High-risk ERCP	1 (2%)
Prior biliary drainage		16 (33%)
Duodenal stenting		10 (21%)
Bilirubin, mg/dL		7.2 (0.4–23.5)
Alkaline phosphatase, U/L		1570 (94–4255)
White blood cell count, µL		7696 (2700–29,200)
C-reactive protein, mg/dL		5.06 (0.10–24.38)
Albumin, g/dL		2.7 (1.4–3.8)

The data are numbers (%) or median (interquartile range). EUS-BD, endoscopic ultrasound-guided biliary drainage; GOO, gastric outlet obstruction; ERCP, endoscopic retrograde cholangiopancreatography.

**Table 2 jcm-10-03111-t002:** Details of the procedure.

Procedure time, min		42 (29–55)
Puncture site	B3	41 (85%)
	B2	7 (15%)
Fistular dilation	Bougie	45 (94%)
	Balloon	40 (83%)
	Cautery	5 (10%)
Stent diameter, mm	8	45 (94%)
	10	3 (6%)
Stent length, cm	10	42 (88%)
	12	6 (13%)

The data are numbers (%) or median (interquartile range).

**Table 3 jcm-10-03111-t003:** Treatment outcomes.

Technical success (*n* = 48)	48 (100%)
Clinical success (*n* = 48)	43 (90%)
Adverse events (*n* = 48)	8 (17%)
Migration (actual and imminent)	5 (1 actual and 4 imminent, 10%)
Peritonitis	3 (6%)
Cholangitis	0 (0%)
Cholecystitis	0 (0%)
Bleeding	0 (0%)

The data are numbers (%).

**Table 4 jcm-10-03111-t004:** Risk factors for stent migration after EUS-HGS.

	Non-Migration Group (*n* = 43)	Migration Group (*n* = 5)	*p*-Value, Univariate	*p*-Value, Multivariate
Age > 70 years, *n* (%)	25, (58.1)	3, (60)	1.000	
Male, *n* (%)	27, (62.8)	3, (60)	1.000	
PS > 1, *n* (%)	27, (62.8)	3, (60)	1.000	
Pancreatic cancer, *n* (%)	21, (48.8)	3, (60)	1.000	
Liver metastasis, *n* (%)	14, (32.6)	3, (60)	0.331	
Ascites, *n* (%)	4, (9.3)	1, (20)	0.637	
Distal stricture, *n* (%)	34, (79)	5, (100)	0.322	
GOO as the reason for EUS-HGS, *n* (%)	23, (53.5)	4, (80)	0.437	
Prior biliary drainage, *n* (%)	13, (30.2)	3, (60)	0.316	
Duodenal stenting, *n* (%)	8, (18.6)	2, (40)	0.276	
Initial distance between the puncture site and the stomach (mm)	20.1 ± 12.8	47.0 ± 11.3	<0.001	0.012
Initial distance from the cardia to the stent (mm)	41.9 ± 14.8	54.0 ± 16.8	0.097	0.099

GOO, gastric outlet obstruction; PS, performance status; EUS-HGS, endoscopic ultrasound-guided hepaticogastrostomy.

**Table 5 jcm-10-03111-t005:** Distance between the stomach and liver at the puncture site before and after EUS-HGS in the migration and non-migration groups.

	Distance between the Stomach and Liver at the Puncture Site		
	Before EUS-HGS	After EUS-HGS	*p*-value
Non-migration group (*n* = 43)	19.1 ± 12.8 mm	17.7 ± 9.8 mm	0.291
Migration group (*n* = 5, 1 actual and 4 imminent)	47.0 ± 11.3 mm	44.8 ± 10.7 mm	0.691

EUS-HGS, endoscopic ultrasound-guided hepaticogastrostomy.

## Data Availability

The datasets generated and analyzed during the current study are not publicly available due to the local privacy policy on clinical data but are available from the corresponding author on reasonable request.

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
