# Peer review of "Risk Factors for Stent Migration into the Abdominal Cavity after Endoscopic Ultrasound-Guided Hepaticogastrostomy"

_jcm, 2021, doi:10.3390/jcm10143111_

Round 1

Reviewer 1 Report

I feel the authors adequately addressed the issues stated in my review. They added some new citations that could enrich the paper and better explain the decisions taken in different scenarios.

Reviewer 2 Report

The paper has been improved. No additional comments.

This manuscript is a resubmission of an earlier submission. The following is a list of the peer review reports and author responses from that submission.

Round 1

Reviewer 1 Report

The manuscript covers the management and prevention of a possible complication after EUS-HGS, which is a relatively new approach and therefore the paper provides un useful information for the further development in this field. Although very infrequent, the abdominal migration is a very serious, potentially fatal complication with difficult solution and the prevention is of utmost importance.

I have some minor comments that could possibly be discussed:

  1. Although the authors offer possible solutions for prevention within the discussion and conclusions, all of them lay within the spectrum of the EUS-HGS. However, once identified the very high risk patients for stent migration, one of the options could be changing the strategy and offer a percutaneous transhepatic biliary drainage (PTBD) instead of assuming the high risk for such complications in current absence of effective prevention. Probably such cases should be discarded EUS-HGS if performed out of highly-specialized centers. Please comment
  2. Some of the patients did not have hyperbilirubinemia (according to the bilirubin range in Table 1). Please comment the indication in those cases
  3. 19% of the patients had hilar obstruction according to Table 1. Such cases often require bilateral drainage and left hepatic duct drainage could not be a complete solution. Please comment

Author Response

    1. Although the authors offer possible solutions for prevention within the discussion and conclusions, all of them lay within the spectrum of the EUS-HGS. However, once identified the very high risk patients for stent migration, one of the options could be changing the strategy and offer a percutaneous transhepatic biliary drainage (PTBD) instead of assuming the high risk for such complications in current absence of effective prevention. Probably such cases should be discarded EUS-HGS if performed out of highly-specialized centers. Please comment

    Response: We are very grateful for your valuable comments despite being busy. We completely agree with this reviewer's opinion, and we think it is better to take alternative treatment such as PTBD when the risk of complications is expected to be high. Therefore, we added the sentence in the discussion as follows. “When preoperative CT clearly indicates long distance between the stomach and liver, it may be an important choice to give up EUS-GBD and treat with alternative methods such as percutaneous transhepatic biliary drainage if there is a high possibility of stent migration.”

    1. Some of the patients did not have hyperbilirubinemia (according to the bilirubin range in Table 1). Please comment the indication in those cases

    Response: Thank you for your question. This cohort includes 33% of cases with prior biliary drainage. These include cases in which a stent is placed via the papilla but the duodenum becomes narrowed due to the progression of the cancer, and EUS-HGS is performed instead of ERCP. Alternatively, a percutaneous catheter was removed and converted to EUS-GBD because the external fistula period was prolonged after PTBD drainage. In these cases, occasionally, bilirubin is not high.

    1. 19% of the patients had hilar obstruction according to Table 1. Such cases often require bilateral drainage and left hepatic duct drainage could not be a complete solution. Please comment

    Response: In our institution, when drainage is necessary not only in the left lobe of the liver but also in the right lobe, an additional stent is placed to bridge the right and left sides so as to cross the hilar, so that drainage of both lobes from the HGS can be performed.

Reviewer 2 Report

Very nice study clearly showing that increased distance from the puncture site to the stomach was associated with the risk of stent migration. I have the following recommendations to strengthen the paper:

  1. Peritonitis should be mentioned in the introduction as a potential complication of this techniques.
  2. Figure 1D is labeled as showing that the distance from the stomach to the puncture site was not different in the stent migration and the non-migration group. This is incorrect since this distance was 47 mm in the migration group and 21 mm in the non-migration group and this was a significant difference. The label on this figure needs to be corrected.
  3. Line 56 lists the  "need for deployment of the proximal end of the stent in the abdominal cavity" as a potential complication. This should be further explained. There is not a need to do this, rather this usually occurs inadvertently.

Author Response

    1. Peritonitis should be mentioned in the introduction as a potential complication of this techniques.

    Response: We, all authors, appreciate the reviewer his/her valuable comments. We added the sentences about peritonitis in the introduction as follows. “Possible complications of EUS-HGS include bleeding, peritonitis, mediastinitis and stent displacement/migration. Stent migration into the abdominal cavity is a serious complication that often requires surgery because it causes severe peritonitis,4, 5 although some stents have been recovered endoscopically6.”

    1. Figure 1D is labeled as showing that the distance from the stomach to the puncture site was not different in the stent migration and the non-migration group. This is incorrect since this distance was 47 mm in the migration group and 21 mm in the non-migration group and this was a significant difference. The label on this figure needs to be corrected.

    Responses: Thank you very much for your precise suggestion. As you pointed out, it was rewritten as follows. “(d) In the stent migration group, the distance between the puncture site and stomach was significantly longer than in the non-migration group.”

    1. Line 56 lists the  "need for deployment of the proximal end of the stent in the abdominal cavity" as a potential complication. This should be further explained. There is not a need to do this, rather this usually occurs inadvertently.

    Response: As you pointed out, it was difficult to convey our intention, so we added and changed the sentence as follows. ”Other complications include an increased distance between the liver and gastric wall (revealing the Candy sign on fluoroscopy) and the opening proximal end of the stent in the abdominal cavity. In such cases, another stent can be placed in tandem to connect the intraabdominal stent to the stomach over the guidewire passed through the deployed stent to prevent bile leakage into the peritoneal cavity7.”